# Variation in Reference Evapotranspiration over the Tibetan Plateau during 1961–2017: Spatiotemporal Variations, Future Trends and Links to Other Climatic Factors

**Yuan Liu, Qianyang Wang, Xiaolei Yao** **, Qi Jiang, Jingshan Yu * and Weiwei Jiang**

College of Water Sciences, Beijing Normal University, Beijing 100875, China;
201921470018@mail.bnu.edu.cn (Y.L.); 201931470001@mail.bnu.edu.cn (Q.W.); yaoxiaolei87@163.com (X.Y.);
201821470010@mail.bnu.edu.cn (Q.J.); 201631470011@mail.bnu.edu.cn (W.J.)
* Correspondence: jingshan@bnu.edu.cn; Tel.: +86-10-5880-7814

**Abstract:** Reference evapotranspiration ($ET_0$) is a key factor in the hydrological cycle and energy cycle. In the context of rapid climate change, studying the dynamic changes in $ET_0$ in the Tibetan Plateau (TP) is of great significance for water resource management in Asian countries. This study uses the Penman–Monteith formula to calculate the daily $ET_0$ of the TP and subsequently uses the Mann–Kendall (MK) test, cumulative anomaly curve, and sliding *t*-test to identify abrupt change points. Morlet wavelet analysis and the Hurst index based on rescaled range analysis (R/S) are utilized to predict the future trends of $ET_0$. The Spearman correlation coefficient is used to explore the relationship between $ET_0$ changes and other climate factors. The results show that the $ET_0$ on the TP exhibited an increasing trend from 1961 to 2017, with the most significant increase occurring in winter; an abrupt change to a tendency to decrease occurred in 1988, and another abrupt change to a tendency to increase occurred in 2005. Spatially, the $ET_0$ of the TP shows an increasing trend from east to west. The change trend of the $ET_0$ on the TP will not be sustainable into the future. In addition, the mean temperature has the greatest impact on the $ET_0$ changes in the TP.

**Keywords:** $ET_0$; abrupt change; Morlet wavelet; Spearman correlation; meteorological elements

## 1. Introduction

As an important factor in the hydrological cycle and atmospheric cycle, evaporation is a key research focus in the field of hydrology and atmospheric science in the context of climate change [1,2]. Because actual evapotranspiration is difficult to predict directly, researchers typically calculate the reference evapotranspiration ($ET_0$) first when calculating the crop-water demand [3–5]. It is important for agricultural management departments to study the long-term variation in trends of $ET_0$ and their relationship to climate factors [6,7].

Reference evapotranspiration, also known as crop reference evapotranspiration, is proposed on the basis of the concept of potential evapotranspiration [8]. Potential evapotranspiration was first proposed by Thornthwaite, which is the maximum evapotranspiration under ideal conditions, including soil evaporation and vegetation evapotranspiration [9]. In the meantime, a large number of scholars, represented by Penman, have continuously improved the theory and method of calculating the potential emission of vegetation [10–14]. Jensen put forward the concept of "crop potential evapotranspiration", which is the upper limit value of evapotranspiration under ideal crop irrigation conditions and introduced the concept of potential evapotranspiration into agriculture for the first time [15]. At present, the widely accepted definition is put forward by FAO (Food and Agriculture

Organization of the United Nations) based on summarizing the research results of Allen et al., which defines reference evapotranspiration ($ET_0$) as the "evapotranspiration rate of the hypothetical crop with a plant height of 12 cm, ground resistance of 70 s/M and albedo of 0.23" [3].

The drastic changes in temperature have significantly affected the trend of evapotranspiration. With the deepening of the research, many scholars have discovered a common phenomenon that as the temperature rises, pan evapotranspiration does not increase simultaneously, but rather shows a downward trend in most regions in the world, so they named it the "evaporation paradox" [16,17]. This phenomenon has been verified in many places around the world since it was proposed. Hobbins et al. found that the actual evapotranspiration in the U.S. has decreased by as much as 64% in the past half-century [18]. The same phenomenon has been found in many parts of India, Australia, and China [19–21]. Some scholars have also found that the phenomenon of the evaporation paradox is obviously regional and seasonal, which makes the research on the trend of evapotranspiration continue to increase [22–24].

Studies show that from the middle and late 20th century to the early 21st century, China's experience of climate change has been substantial in that there are abrupt climate changes and obvious trends in most areas of China [25–28]. The Tibetan Plateau (TP) is not only the main source of rivers in China and Asia, but also an important ecological barrier that is highly sensitive to global climate change [29–31]. The TP is a unique alpine mountain ecosystem that has had an intense response to climate change, especially in agriculture and animal husbandry [32–34]. More importantly, the water budget and energy exchange in the region have a considerable impact on the intensity of the Asian monsoon and high pressure [35,36]. Guo et al. concluded, by studying the trend of the evapotranspiration index of standard precipitation in China, that winter and spring in the TP will become drier in the 21st century [37]. Ye et al. further found that the frequency of extreme drought events in the Qaidam Basin is the highest (3–6 times) in the TP [38]. Wang et al. considered that drought is caused by a decrease in precipitation and an increase in temperature [39]. Some scholars have also found that the TP has experienced a continuous wet trend since the 1950s [40–44]. The prolonged rainy season and increased precipitation can effectively alleviate drought in the region [41,43]. Different conclusions can be drawn from the study of different drought indexes throughout different time ranges.

In the past half century, the evapotranspiration in the TP has experienced several significant transformations (1–3 times) [43,45,46]. At present, the Mann–Kendall (MK) test, Theil–Sen estimate, linear trend, sliding *t*-test, Pettitt test, and cumulative anomaly curve are commonly used to study the trend and changes of climate elements. Jiang et al. used the MK test, sliding *t* test, cumulative anomaly curve, and other methods to identify the trends and change-points of the TP runoff [47]. Attarod et al. used the MK test to examine the changing trends of meteorological parameters and $ET_0$ in the Zagros region of western Iran [48]. Sonmez and Kale used the Pettitt analysis to determine that the hydrometeorological elements of the Filyos River (Turkey) experienced abrupt changes in approximately 2000 [49]. Chu et al. used the MK test and the Theil–Sen estimate to determine that the abrupt change point of $ET_0$ in the Huaihe River Basin was approximately 1990 [50]. Wang et al. used the MK test and the Pettitt test change-point statistics to find that the hydrometeorological elements of the Taihang Mountain area changed abruptly in approximately 1982 [51]. Similarly, Liu et al. used the MK test and moving *t*-test methods to find that the evapotranspiration in the source region of the Yellow River on the TP had a sudden change during the 1980s [52]. However, a single method is typically inadequate for the identification of the abrupt change points of hydrometeorological elements, and it is necessary to integrate multiple methods to determine the abrupt change points of a certain sequence [53]. With regards to exploring the causes of changes in $ET_0$ on the TP, scholars have arrived at different conclusions using varying research approaches. Liu et al. analyzed the meteorological elements in the active layer of the TP during different freeze-thaw stages and found that the contribution of each meteorological element to $ET_0$ varied by season [54]. Therefore, with the extension of the research period and the increase in the number of factors considered, it is necessary to continue to explore the relationship between $ET_0$ and other meteorological elements.

In summary, this study attempts to answer the following three questions: (1) What are the temporal and spatial trends of $ET_0$ in different seasons across the TP from 1961 to 2017? (2) Does this time change trend have periodicity, or will such changes in trends continue into the future? (3) What is the relationship between $ET_0$ and various meteorological elements and energy conditions, and which elements play a key role in this relationship? These results will deepen our understanding of climate change and guide the formulation of river drought warning and crop irrigation plans in many Asian countries.

## 2. Materials and Methods

### 2.1. Study Area and Data

The Tibetan Plateau (26°00′~39°47′ N, 73°19′~104°47′ E) is an Asian inland plateau with a total area of approximately 2.5 million square kilometers and an average altitude over 4000 m (Figure 1). The TP is the highest plateau in the world and is known as the "Third Pole" and the "Roof of the World" [55,56]. As the most sensitive area in the world to global climate change, the TP has been warming at a rate approximately twice that of global warming in the past 50 years [57]. Luo et al. found that the number and area of glacial lakes has increased by 56% and 35%, respectively [58]. This warming has seriously threatened the glaciers and freshwater reserves in the region and will affect the water supply security of billions of people [56,57,59].

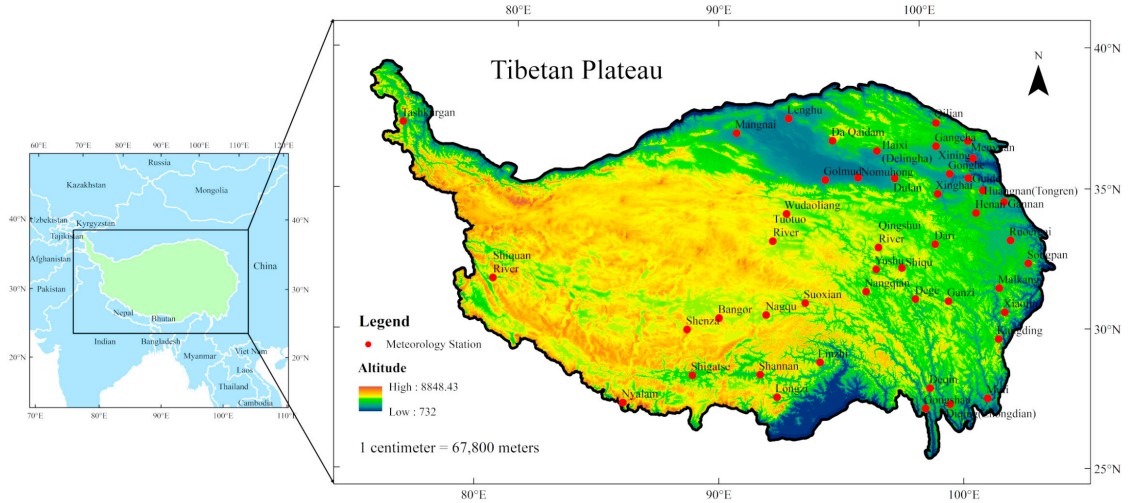

**Figure 1.** The location of the Qinghai-Tibet Plateau and the distribution of meteorological stations.

Forty-six meteorological stations in the TP were selected to analyze the temporal and spatial evolution of sequences (Figure 1). Climate data from 1961 to 2017, including daily mean temperature (Tmean,°C), maximum temperature (Tmax, °C), minimum temperature (Tmin, °C), precipitation (P, mm), relative humidity (RH, %), actual atmospheric pressure (AP, kPa), sunshine duration (SD, hour), and wind speed (U₂, m/s), were obtained from the China Meteorological Data Service Center (CMDC) (http://data.cma.cn/).The dataset is repeatedly monitored and controlled before it is released and has good quality. In addition, for the few missing data, we interpolate the missing data by referring to the observation data of the same period with similar topographical conditions and the altitude of the missing station.

### 2.2. Analytical Methods

#### 2.2.1. FAO Penman–Monteith Formula

The Penman–Monteith (P–M) formula is the only standard method recommended by the FAO to calculate $ET_0$ and is currently the most widely used method [3,60]. In this paper, the P–M formula is used to calculate daily $ET_0$ values:

$$ET_0 = \frac{0.408\Delta(R_n - G) + \gamma \frac{900}{T_{mean}+273}U_2(e_s - e_a)}{\Delta + \gamma(1 + 0.34U_2)} \tag{1}$$

where $\Delta$ is the slope of the relationship curve between the saturated water pressure and temperature (k Pa°C$^{-1}$); $R_n$ is the net radiation on the crop surface (MJ m$^{-2}$·day$^{-1}$); $G$ is the soil heat flux (MJ m$^{-2}$·day$^{-1}$); $\gamma$ is the hygrometer constant (k Pa°C$^{-1}$); $T_{mean}$ is the average air temperature (°C); $U_2$ is the wind speed at a height of 2 m above the ground (m s$^{-1}$); $e_s$ is the air saturation vapor pressure (kPa); and $e_a$ is the actual vapor pressure (k Pa).

The net radiation was calculated using the following formula:

$$R_n = R_{ns} - R_{nl} \tag{2}$$

$$R_{ns} = (1 - \alpha)R_s \tag{3}$$

$$R_s = \left(a_s + b_s\frac{n}{N}\right)R_a \tag{4}$$

$$R_a = \frac{24(60)}{\pi}G_{sc}d_r[w_s \sin\varphi \sin\theta + \cos\varphi \cos\theta \sin w_s] \tag{5}$$

$$R_{nl} = \sigma\left[\frac{T^4_{max,k} + T^4_{min,k}}{2}\right]\left(0.34 - 0.14\sqrt{e_a}\right)\left(1.35\frac{R_s}{R_{so}} - 0.35\right) \tag{6}$$

where $R_{ns}$ is shortwave radiation (MJ m$^{-2}$·day$^{-1}$); $\alpha$ is the canopy reflectance of vegetation with a value of 0.23 [3]; $n$ is the sunshine duration (h); $N$ is the maximum possible sunshine duration (h); $R_a$ is the solar radiation at the top of the atmosphere (MJ m$^{-2}$·day$^{-1}$); $G_{sc}$ is the solar constant with a value of 0.082 (MJ m$^{-2}$min$^{-1}$); $d_r$ is the inverse relative distance between the Earth and Sun; $w_s$ is the angle of sunset sun (rad); $\varphi$ is the latitude (rad); $\theta$ is the solar declination (rad); $R_{nl}$ is the net longwave radiation (MJ m$^{-2}$·day$^{-1}$); $\sigma$ is the Stephen–Boltzmann constant, with a value of $4.903 \times 10^{-9}$ (MJ K$^{-4}$ m$^{-2}$day$^{-1}$); $T_{max,k}$ is the maximum daily air temperature (K); $T_{min,k}$ is the minimum daily air temperature (K); and $R_{so}$ is the value for short wave radiation on the surface of vegetation on sunny days (MJ m$^{-2}$·day$^{-1}$). Ye et al. found that $a_s$ = 0.24, $b_s$ = 0.6 and $R_{so} = \left(0.64 + 5.48 \times 10^{-5}Z\right)R_a$ are suitable for TP, where $Z$ is the altitude (m) [61].

#### 2.2.2. Cumulative Anomaly Curve

A cumulative anomaly curve can indicate the long-term evolution trend and continuous change of a sequence, and it is a common method for determining the change trend of hydrological and meteorological sequences [62,63]. The method for calculation of the cumulative anomaly is as follows:

$$\overline{x_t} = \sum_{i=1}^{t}(x_i - \overline{x}), t = 1, 2, 3 \ldots n \tag{7}$$

$$\overline{x} = \frac{1}{n}\sum_{i=1}^{n}x_i, i = 1, 2, 3 \ldots n \tag{8}$$

where $x_i$ is the meteorological dataset and $n$ is the length of the dataset. The anomaly is the distance to the average value, which reflects the degree of the data dispersion. An increase in the cumulative anomaly curve indicates an upward trend, while a decrease indicates a downward trend. Since the abrupt change point must appear near the peak or inflection point of the cumulative anomaly curve, the year of the change can be approximated.

### 2.2.3. Mann–Kendall Test

As recommended by the WMO, the Mann–Kendall (MK) test is widely used as a nonparametric test of temperature, precipitation, and other factors [64]. The advantage of this test is that the samples do not need to follow a particular distribution and are not disturbed by small fluctuations. The MK test can reveal the trend and transformation of a dataset [65].

The definitions of the two statistics, S and Z, and the calculation process are as follows:

$$S = \sum_{i=1}^{n-1} \sum_{j=i+1}^{n} sgn(x_j - x_i) \tag{9}$$

$$sgn(\theta) = \begin{cases} 1, & if\ \theta > 0 \\ 0, & if\ \theta = 0 \\ -1, & if\ \theta < 0 \end{cases} \tag{10}$$

$$Var(s) = \frac{n(n-1)(2n+5)}{18} \tag{11}$$

$$Z = \begin{cases} \frac{s-1}{\sqrt{var(s)}}, & s > 0 \\ 0, & \theta = 0 \\ \frac{s+1}{\sqrt{var(s)}}, & s < 0 \end{cases} \tag{12}$$

where $x_i$ and $x_j$ are two sequential values in the meteorological dataset ($1 < i < j < n$); $n$ is the size of the dataset; sgn is a symbolic function; and Z is an indicator of the severity of the change trend. A positive value indicates that the trend is rising, while a negative value indicates that the trend is falling. When $|Z| \geq |Z_{(1-\alpha)/2}|$, the data series can be said to change significantly at the $\alpha$ significance level.

$d_k$ is defined as the sum of the cumulative numbers when $x_j > x_i$ ($1 < i < j < n$). The mathematical expectation and variance of $d_k$ are as follows:

$$E(d_k) = \frac{n(n-1)}{4} \tag{13}$$

$$D(d_k) = \frac{k(k-1)(2k+5)}{72} \tag{14}$$

$$UF_k = \frac{d_k - E(d_k)}{\sqrt{D(d_k)}} \tag{15}$$

where $2 \leq k \leq n$ and $UF_1 = 0$; if $UF_k \geq |U_{\alpha/2}|$, the data series can be said to change significantly at the $\alpha$ significance level. $UD_k$ is the negative sequence of $UF_k$. The MK mutation chart is to draw $UF_k$, $UD_k$, and the confidence interval in the same line chart. This chart has the following three characteristics:

① When $UF_k > 0$, it means that the sequence shows an upward trend, otherwise, it is a downward trend.
② When the $UF_k$ line exceeds the critical line, it indicates a significant change trend.
③ When the intersection of the $UF_k$ curve and the $UD_k$ curve is within the confidence interval, the intersection point is a mutation point.

### 2.2.4. Other Methods

In order to determine the location of the abrupt change point more accurately, the moving *t*-test method is used in addition to the cumulative anomaly curve and the MK test. Wavelet analysis and the Hurst index analysis based on R/S are used to determine the change period and future trend [66]. Spearman's correlation analysis is used to explore the relationship between $ET_0$ and other factors [67].

## 3. Results

### *3.1. The Temporal Trend of $ET_0$*

Using the Thiessen polygon method to calculate the regional average $ET_0$ value, and through the MK trend test ($\alpha = 0.05$), the statistical value $Z$ of the annual and seasonal $ET_0$ from 1961 to 2017 is calculated (Table 1). It shows that the Z value of the annual $ET_0$ sequence was 1.59, which indicates that the $ET_0$ in the TP was increasing, but the trend is not significant. The change trends of spring and autumn are the same, with a significant increasing trend in winter ($\alpha = 0.001$) and an insignificant downward trend only in summer. The cumulative anomaly method, the MK test and the 3-step sliding *t*-test are used to identify the abrupt change point (Figure 2). Figure 2a shows that the positive peak value of the cumulative anomaly curve occurred in 1988, indicating that the $ET_0$ experienced a sudden change towards a tendency to decrease, and that, after 2005, the negative peak value tended to increase. As shown in Figure 2b, the intersection points of the positive and inverse sequence lines of the MK test in the confidence interval are in 1963, 1984, and 2013. Because the sequences before 1963 and after 2013 are too short to permit the use of these years as the abrupt change point, 1984 is considered the year of transition. In Figure 2c, the positive peaks of the *t*-test curves of the 5-year, 10-year, and 15-year sliding step lengths that exceeded the confidence interval all occurred in 1988, and the negative peaks beyond the lower critical line appeared in 2005. These results are consistent with those of the cumulative anomaly method. Based on the three test methods, it is determined that the transitional years for the $ET_0$ sequence from 1961 to 2017 were 1988 (beginning of decreasing trend) and 2005 (beginning of increasing trend).

**Table 1.** Z-values from the Mann–Kendall (MK) test of reference evapotranspiration $(ET_0)$ at different scales.

| Scale | Year | Spring | Summer | Autumn | Winter |
|---|---|---|---|---|---|
| Z value | 1.59 | 1.58 | −1.08 | 0.86 | 3.93 *** |

*** Significance level of 0.001 ($p < 0.001$).

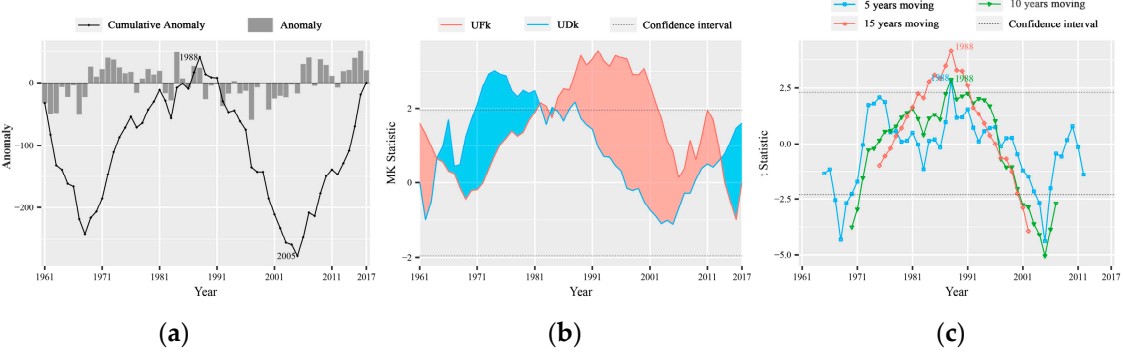

(a)                                             (b)                                             (c)

**Figure 2.** The abrupt change point identification diagram of the reference evapotranspiration ($ET_0$) sequence, where (**a**) is the cumulative anomaly curve; (**b**) is the positive and negative sequence line of the Mann–Kendall (MK) test; and (**c**) is the sliding *t*-test curve with three steps, and the corresponding color is marked as the peak value of each curve.

The results of the $ET_0$ change analysis in the four seasons are shown in additional information (Figures 3–6). Figure 3b shows that the intersections of the spring MK test curve in the confidence interval occurred in 1963, 1981, 1983, 1985, 2003, 2004, and 2005. Among these years, 2003 was the negative peak of the cumulative anomaly curve, after which the $ET_0$ tended to increase; 1981 and 1985 were the abrupt change points of the cumulative anomaly curve after which the $ET_0$ tended to decrease. Figure 4 shows that the three test results of the summer $ET_0$ sequence were relatively consistent. There was only one intersection point in the middle of the MK curve, which was 1991. Moreover, 1991 was the positive peak of the cumulative anomaly curve, which also exceeded the upper critical line in the *t*-tests with 10-year and 15-year sliding step lengths, indicating that there was a change that tended to decrease at this time. Figure 5 shows that the MK test of the autumn $ET_0$ did not identify effective abrupt change points, and the cumulative anomaly curve showed a positive peak in 1988, which also exceeded the upper critical line in the 10-year and 15-year sliding step *t*-tests. The autumn series shows a decreasing abrupt change in 1988, which was consistent with the results of the annual series. Figure 6 shows that the period from 1999 to 2004 was the negative peak section of the cumulative anomaly curve for the winter $ET_0$ series, the 5-year and 10-year step sliding *t*-test curves exceeded the lower critical line in 2004, and the winter $ET_0$ exhibited a sudden change that tended to increase in 2004.

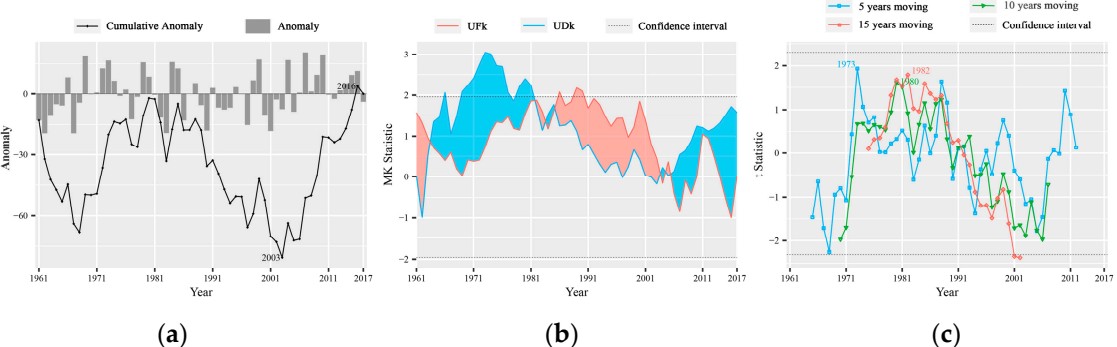

**Figure 3.** The abrupt change point identification diagram of spring $ET_0$ sequence, where the picture category of the (**a**–**c**) is same to Figure 2.

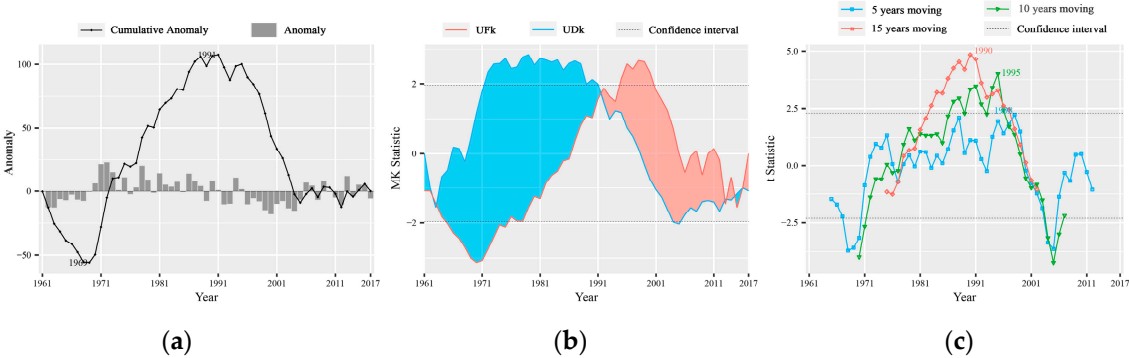

**Figure 4.** The abrupt change point identification diagram of summer $ET_0$ sequence, where the picture category of the (**a**–**c**) is same to Figure 2.

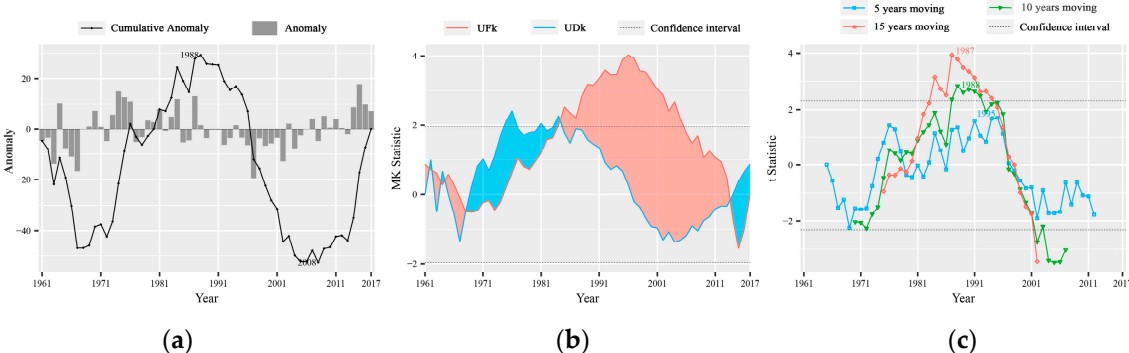

**Figure 5.** The abrupt change point identification diagram of autumn $ET_0$ sequence, where the picture category of the (**a**–**c**) is same to Figure 2.

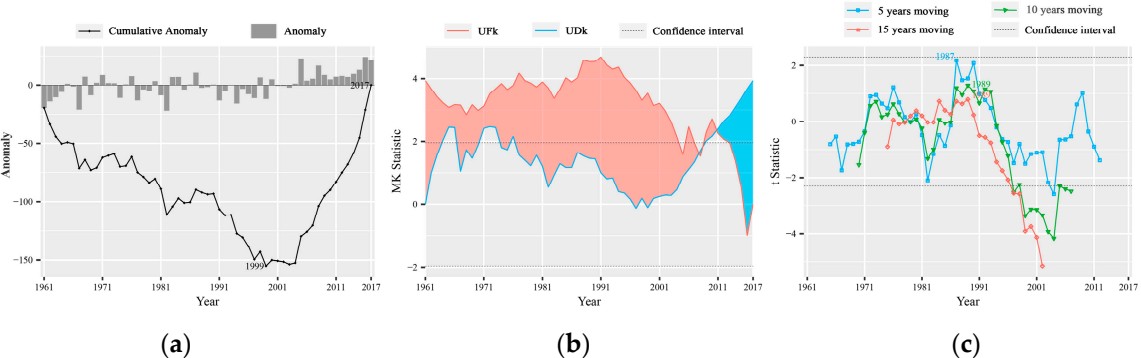

**Figure 6.** The abrupt change point identification diagram of winter $ET_0$ sequence, where the picture category of the (**a**–**c**) is same to Figure 2.

### 3.2. The Spatial Distribution of $ET_0$

Figure 7 shows the spatial distribution trend of the annual average $ET_0$ of the TP from 1961 to 2017 and the MK test statistical value Z of each station. Overall, the annual average $ET_0$ of the TP gradually increased from east to west, and local peak centers appeared in the northern Qaidam Basin, the southeastern Himalayas, and the Hengduan Mountains. For the distribution of the Z values, 69.56% of the stations increased and 65.62% increased significantly, but the Qaidam Basin area decreased.

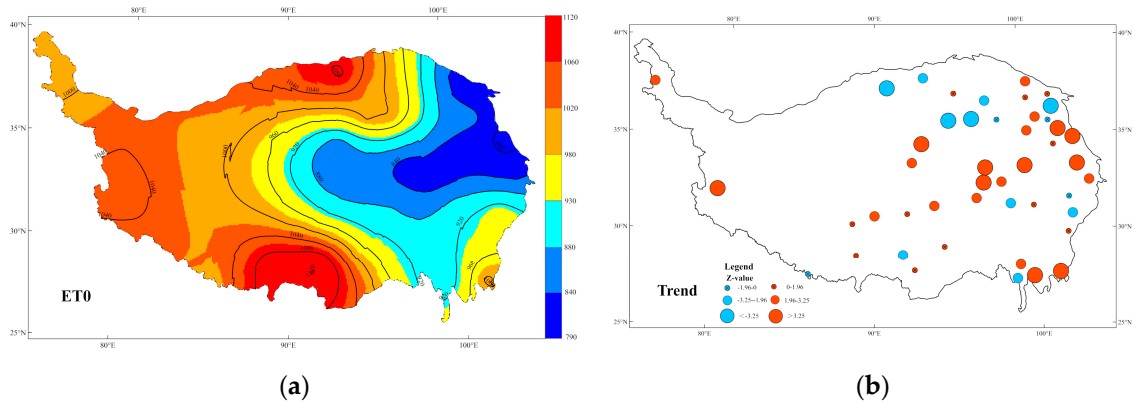

**Figure 7.** (**a**) The distribution of the multi-year average $ET_0$ in the Tibetan Plateau (TP); (**b**) the distribution of the statistical value Z of the Mann–Kendall test at each station of TP from 1961 to 2017. At the $\alpha$ = 0.05 significance level, if |Z| > 1.96, $ET_0$ changes significantly. The red bubble represents the increasing trend, the blue bubble represents the decreasing trend, and a larger bubble size suggests an increased significance.

The spatial distribution of the $ET_0$ in spring, summer, and autumn in the TP was the same as that of the annual $ET_0$, showing an increasing trend from the east to the west (Figure 8). However, there was an increasing trend from the northeast to the southwest in winter (Figure 8d). Winter presented an increasing trend from the northeast to the southwest (Figure 8). In addition, the region had the highest average $ET_0$ in spring and the lowest in winter. The contour of the $ET_0$ was the densest in summer, which shows that the spatial variation is the most dramatic in this season (Figure 8b).

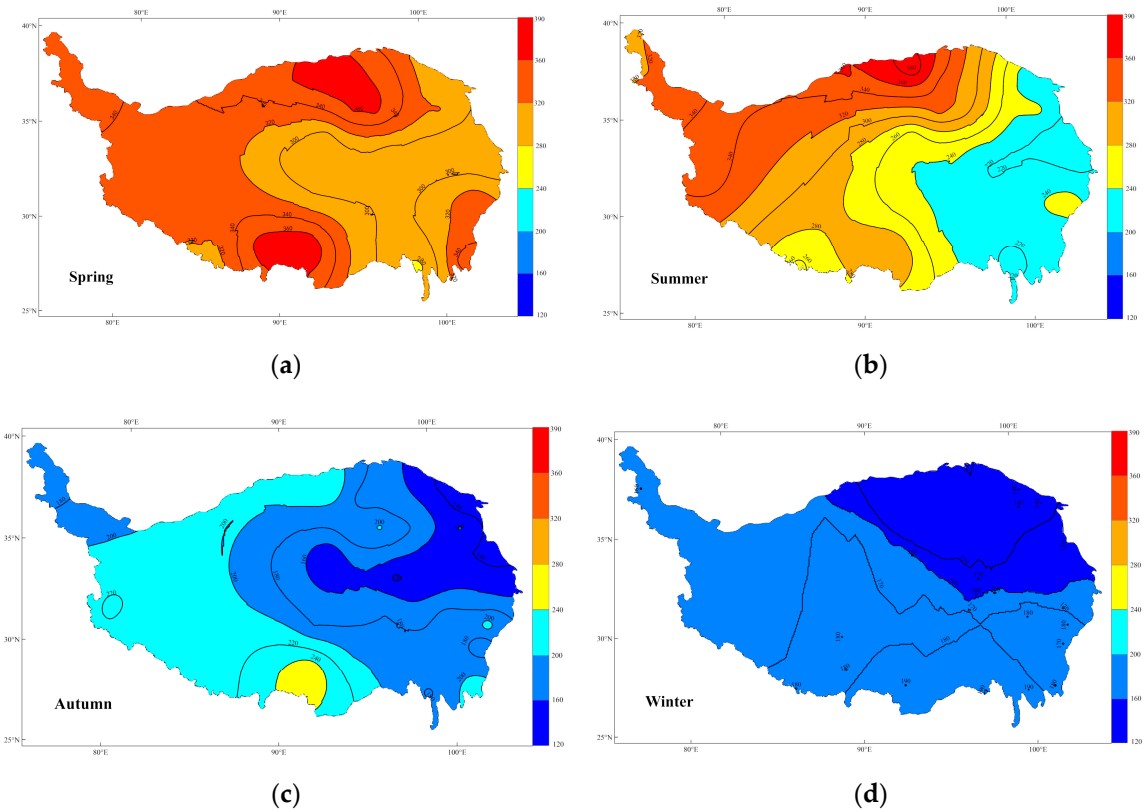

**Figure 8.** The spatial distribution of $ET_0$ in the four seasons, where (**a**) is spring; (**b**) is summer; (**c**) is autumn; (**d**) is winter.

In the four seasons, the variation trend of the $ET_0$ in spring was relatively mild, and the number of stations with a significant trend ($|Z| > 1.96$) accounted for only 45.65% of stations (Figure 9a), but the increasing trend of the $ET_0$ at the two stations in the western TP was the most significant in spring. The distribution patterns in summer and autumn were similar, the decreasing trend in the Qaidam Basin was more significant in summer (Figure 9b), and the increasing trend in the TP was more obvious in autumn (Figure 9c). The increasing trend of the $ET_0$ was the most significant in winter, with 89.13% of stations increasing, of which 73.91% increased significantly ($Z > 1.96$), and 26.09% of the stations increased very significantly ($Z > 3.25$). The number of sites showing an increasing trend gradually increased, and the increasing trend accelerated from winter to spring.

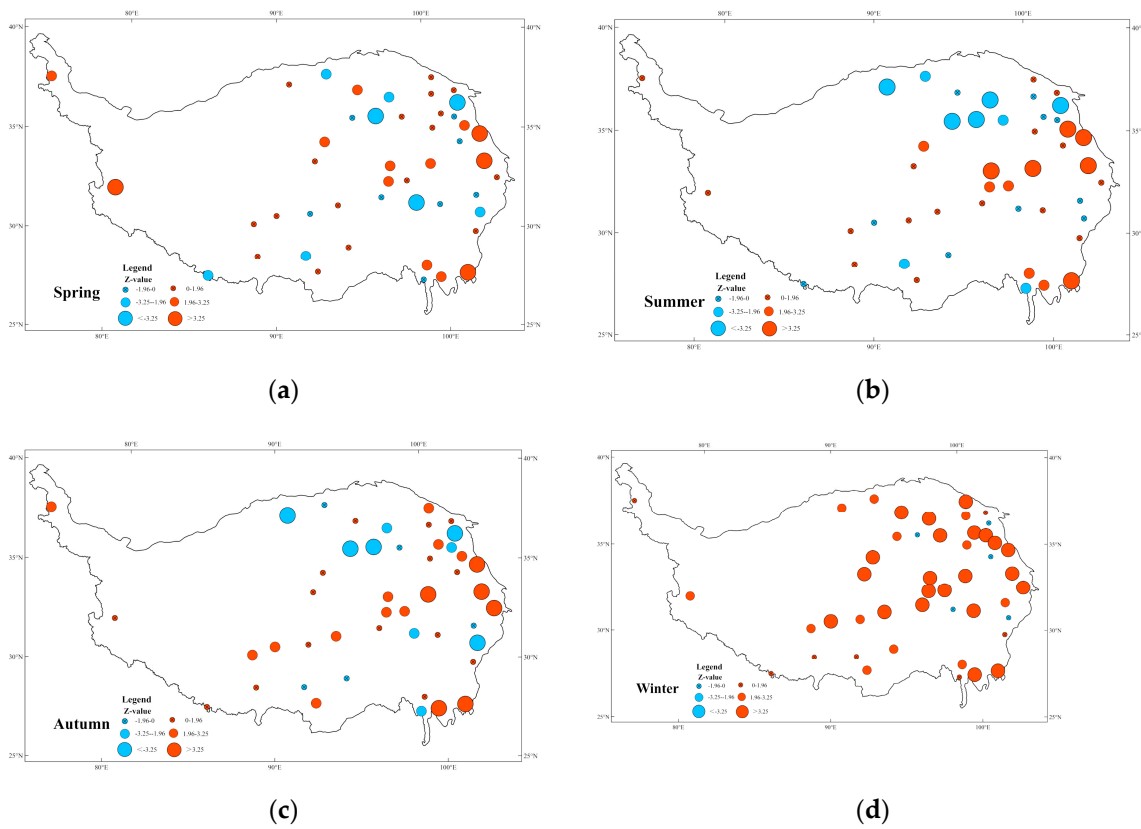

**Figure 9.** The distribution of the of the Mann–Kendall Z statistic for $ET_0$ series at each station of from 1961 to 2017, where (**a**) is spring; (**b**) is summer; (**c**) is autumn; (**d**) is winter. The significance level and expression method are the same as in Figure 7b.

### 3.3. The Future Trend of $ET_0$

After reducing the boundary effect, the complex Morlet wavelet transformation was used to analyze the $ET_0$ series. The isoline map of the real part of the wavelet analysis (Figure 10) and the time-frequency distribution map of the wavelet transform modulus (Figure 11) were plotted. Figure 10 shows that the 24–30a scale is very prominent, and that its scale center is approximately 27a. On this time scale, there are four partial high value centers and three low value centers in the $ET_0$ of the TP, which occurred in 1962, 1981, 1999, and 2017 and 1973, 1989, and 2009, respectively. Figure 11 shows the oscillation energy in different periods. The oscillation energy is the strongest, and the period is the most significant, for the 20–32a time scale, but its periodic change is local (before 1973 and after 2005).

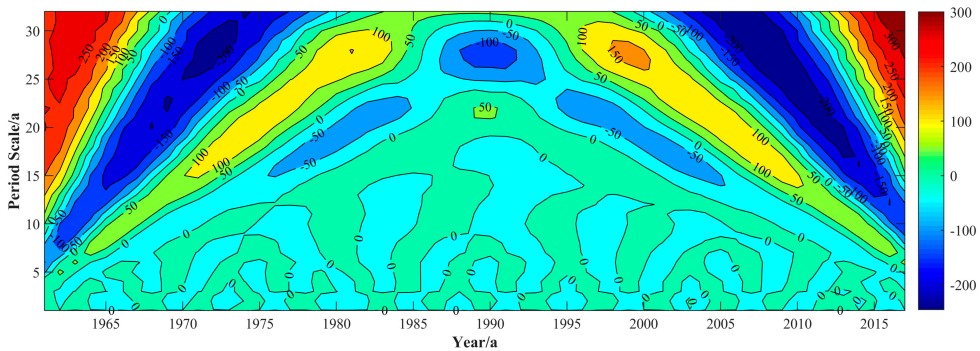

**Figure 10.** Contour map of the real part of wavelet coefficients.

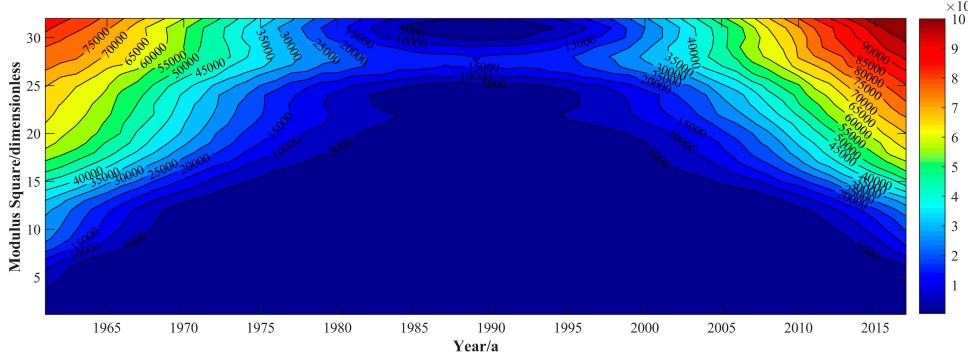

**Figure 11.** Time-frequency distribution of the wavelet transform modulus.

Figure 12a shows the wavelet variance diagram of the annual $ET_0$ series over the TP. There is only one main peak value of the wavelet variance corresponding to the 28a time scale, which indicates that the $ET_0$ sequence of the TP from 1961 to 2017 had the strongest periodic oscillation of approximately 28a, and the variance curve finally showed an upward trend, suggesting that it is likely that there will be another extreme value, indicating that there may be major periodic changes greater than 32a. Figure 12b shows that the average annual $ET_0$ variation period of the TP is approximately 10 years on a 28a time scale, and the TP experienced approximately three wet–dry transition periods.

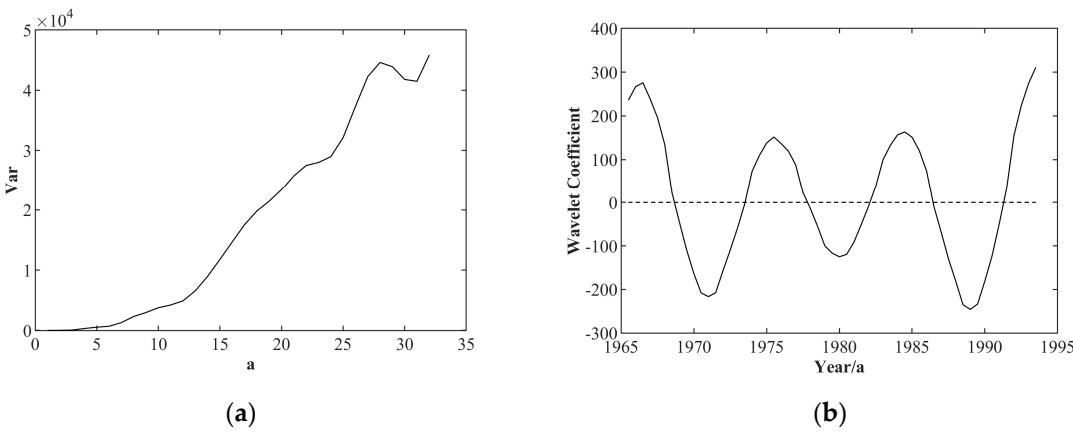

(**a**)  (**b**)

**Figure 12.** (**a**) The wavelet variance analysis chart; (**b**) the wavelet coefficient chart for the 28a time scale.

Table 2 shows the Hurst index of the $ET_0$ sequence of the year and the four seasons. The Hurst index of the $ET_0$ sequence of each time scale, calculated by the R/S analysis method, are all less than 0.5, which suggests that the future trend is anti-continuous.

**Table 2.** The Hurst index of the $ET_0$ series for the year and four seasons.

| Scale | Year | Spring | Summer | Autumn | Winter |
|---|---|---|---|---|---|
| Hurst index | 0.31 | 0.21 | 0.28 | 0.30 | 0.31 |

### 3.4. The Relationships between the $ET_0$ and Meteorological Elements and Energy Conditions

In this part, we try to explore the main factors that affect $ET_0$ changes in different regions by comparing the correlation coefficient between the $ET_0$ and each factor. The Spearman analysis has been used to calculate the correlations between the $ET_0$ and RH, P, Tmean, Rn, SD, and $U_2$ separately. Figure 13a shows the Spearman correlation coefficients between the $ET_0$ changes of each meteorological station on the TP between 1961 and 2017 as well as other meteorological elements. There is a negative correlation between RH and P and the $ET_0$. Tmean, Rn, SD, $U_2$, and the $ET_0$ in most regions were

positively correlated. Among them, Tmean has had a greater impact on the central and eastern TP, Rn has had a greater impact on the central and southern regions, SD has had a greater impact on the east, and $U_2$ has had a greater impact on the north. In addition, RH, Tmean, and $U_2$ has had the greatest impacts on $ET_0$.

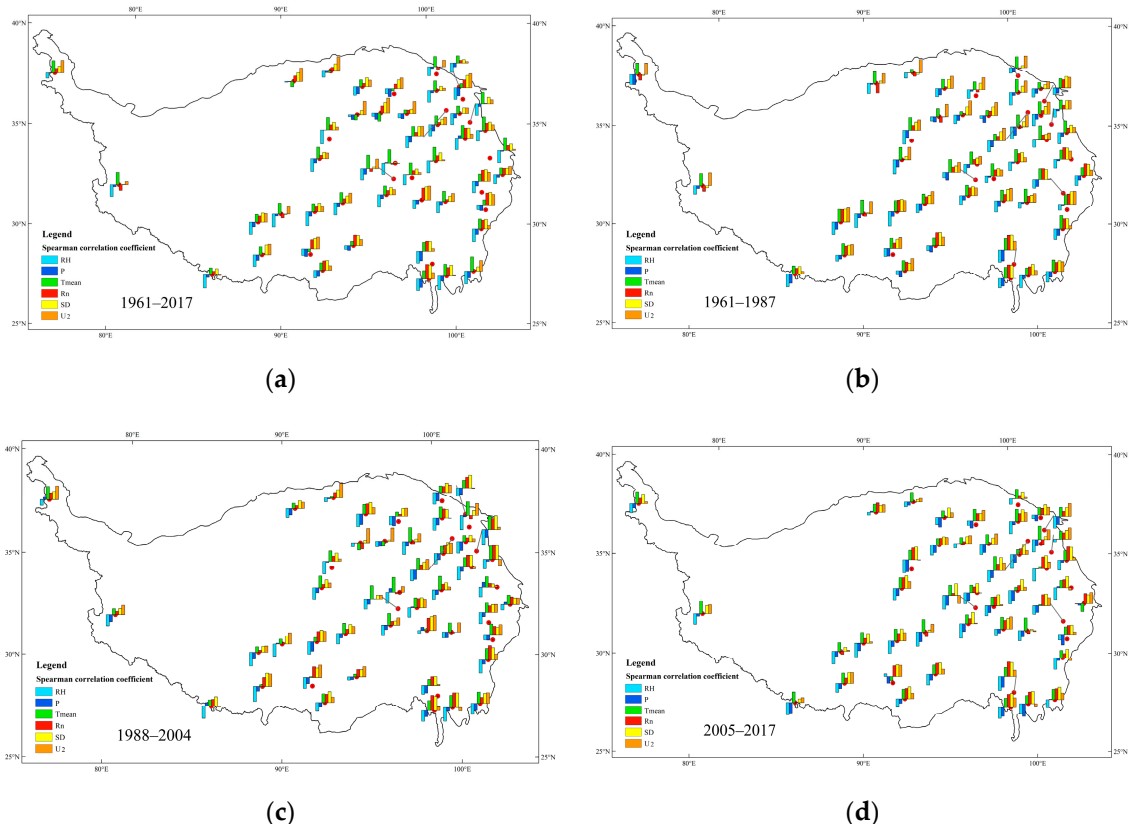

**Figure 13.** Distribution of correlation between the $ET_0$ and other factors; (**a**) 1961–2017, (**b**) before the first abrupt change (1961–1987); (**c**) between two abrupt changes (1988–2004); (**d**) after the second abrupt change (2005–2017).

The results in Section 3.1 show that a tendency to decrease occurred in the annual $ET_0$ sequence of the whole region beginning in 1988, and a tendency to increase occurred beginning in 2005. The relationships between the $ET_0$ series of these three periods and other meteorological elements were studied separately. The comparison of Figure 13b,c shows that before the first abrupt change (1961–1987), the influencing factors that had a positive correlation with the $ET_0$ had a greater impact, especially Tmean, which had a significant impact on the entire region, and SD and $U_2$ had a greater impact on the southern TP. After the abrupt change in 1988, the influence of RH, which had a negative correlation with the $ET_0$, increased dramatically throughout the whole region, and the influence of Rn on the eastern region also increased. Figure 13c,d shows that after the sudden increase in 2005, the influence of RH and P on the central region increased, and the influence of $U_2$, which has a positive correlation with the $ET_0$, on the northeastern TP increased.

## 4. Discussion

### 4.1. Variation of ET₀ Sequence

This study concludes that the $ET_0$ of the TP showed an increasing trend between 1961 and 2017, with a particularly significant ($\alpha = 0.001$) increase in $ET_0$ occurring in winter. This is inconsistent with the phenomenon of the evaporation paradox previously discussed by many scholars. Zhang et al. and

Liu et al. studied pan evaporation from the 1960s to the beginning of the 21st century and found that the TP also had an evaporation paradox [68,69]. This was caused by the inconsistent length of the time series selected in the study. The longer time series selected in this study was closer to the current changes. Similarly, the results obtained by some scholars, including those after 2005, are also find that pan evapotranspiration experienced an initial decrease, followed by an increase [70–72], which is consistent with those outlined here. Fan et al. also came to this conclusion that the increasing trend in the spring and winter seasons was more significant [71].

*4.2. Abrupt Change in the $ET_0$ Sequence*

Similarly, the length of the time series selected also has a decisive influence on the results of change analysis. This study fully considered the limitations of the research method when conducting climate change research, and therefore, three methods, the MK test, cumulative anomaly method, and sliding *t*-test, were used to identify the abrupt change points. This study concludes that for the annual $ET_0$ sequence that occurred between 1961 and 2017, there was a tendency to decrease beginning in 1988 and a tendency to increase beginning in 2005. Many scholars also concluded that abrupt changes occurred in the 1980s [45,46,51,52]. The increasing trend at the beginning of the 21st century is reported less frequently in the existing conclusions, but it was also noted by previous scholars [45]. This is because, in previous studies, the abrupt change points at the beginning of the 21st century were closer to the end of the sequence, making it impossible to draw conclusions. In this study, a time series with a length of 12 years after the abrupt change point in 2005 was detected, allowing it to be characterized as an abrupt change point. The time series selected in this paper is long enough to prove this phenomenon.

*4.3. The Linkages of $ET_0$ with Other Factors*

Differing from the Pearson correlation coefficient, the Spearman correlation coefficient chosen for this study does not make assumptions about the distribution of variables and is suitable for non-parametric tests [73]. Our research found that the change in the $ET_0$ between 1961 and 2017 had the greatest positive correlation with Tmean, followed by $U_2$ and SD. Jian et al. similarly found an increase in the $ET_0$, due to temperature changes [74]. Chen et al. and Xie and Zhu concurred that wind speed has an important influence on the $ET_0$ of the TP [75,76]. Ma et al. found that energy conditions are a key factor and are also related to the SD obtained by this research [77]. Song et al. found that the decline in the ET on the TP from 2000 to 2010 could be attributed to relative humidity [78], in our research results, the $ET_0$ also shows a strong correlation with relative humidity in this period. To further explore the differences in influencing factors in different regions, the Spearman coefficient of each site was spatially plotted. The influence on sites in high-altitude areas in southwestern China was lower for Tmean than for Rn, SD, and $U_2$. This could be attributed to the thinner air in high-altitude areas, which is affected by energy conditions.

## 5. Conclusions

Based on the meteorological data measured at 46 meteorological stations on the TP between 1961 and 2017, this study calculated daily net solar radiation and $ET_0$ using the Penman–Monteith formula. Compared with most studies, the period of this study is larger, the rules obtained are more accurate, and the research results are more convincing. In addition, to explore the time trend of $ET_0$, the MK test, cumulative anomaly method, and sliding *t*-test were used to identify abrupt change points. Periodicity was explored through the Morlet wavelet analysis and Hurst index based on R/S analysis. In addition, the Spearman correlation coefficient was utilized to explore the relationship between $ET_0$ and other meteorological elements. The main conclusions of this research are as follows:

(1)　　The annual $ET_0$ of the TP showed an increasing trend from 1961 to 2017, where the increasing trend of the $ET_0$ in winter was particularly significant. The $ET_0$ sequence experienced two abrupt changes. The first occurred in 1988, to initiate a decreasing trend, and the second occurred in 2005, to initiate an increasing trend.

(2)　　The multiyear averaged $ET_0$ sequence of the TP showed an increasing distribution from east to the west in space. Among the four seasons, spring showed the highest $ET_0$, and the contours of $ET_0$ were the densest in summer, with the most dramatic spatial changes. Three seasons, spring, summer and autumn showed a low $ET_0$ in the east and a high $ET_0$ in the west, but in winter, the $ET_0$ was high in the south and low in the north.

(3)　　The first main cycle of the annual $ET_0$ in the Tibetan Plateau appeared on a 28-year time scale, and we observed three dry and wet cycles at 10-year intervals. Because the Hurst index is less than 0.5 (0.31), the future change trend of the annual $ET_0$ series in the TP is anti-persistent.

(4)　　Regarding changes in the annual $ET_0$ sequence of the TP between 1961 and 2017, the positively correlated factor Tmean played a leading role, followed by $U_2$ and SD, and the negatively correlated factor RH had a great impact. In addition, Rn and SD had a greater impact in high-altitude areas.

**Author Contributions:** Conceptualization, methodology, software, Y.L.; validation, X.Y. and Q.W.; formal analysis, writing—original draft preparation, Y.L. and Q.W.; investigation, resources, data curation, W.J. and Q.J.; supervision, project administration, funding acquisition, J.Y. All authors have participated in the review and editing. All authors have read and agreed to the published version of the manuscript.

**Funding:** This research was funded by the National Natural Science Foundation of China, grant number 51779007 and the National Key Research and Development Program of China, grant number 2016YFC0401308.

**Acknowledgments:** The authors would like to thank the China Meteorological Data Service Center (CMDC) for providing data. The authors are very grateful to the four reviewers for their professional advice, which not only improves the details and methods of this paper, but also opens up the research ideas for the next stage.

**Conflicts of Interest:** The authors declare no conflict of interest.

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
