# Peer review of "Variation in Reference Evapotranspiration over the Tibetan Plateau during 1961–2017: Spatiotemporal Variations, Future Trends and Links to Other Climatic Factors"

_water, doi:10.3390/w12113178_

Round 1
Reviewer 1 Report
In my opinion, the work is well organized and based on solid and well-known techniques. Furthermore, the results are clearly presented and they accomplish the objectives of the work.
However, some minor issues need to be revised:
- In the Introduction, please, define the concept of "reference evapotranspiration" and how it differs from potential evapotranspiration.
- Please, explain all the acronyms before mentioning them. Abstract (lines 18 "MK" and 19 "R/S"), Line 108 (before "P-M" add "Penman-Monteith").
- Lines 103-105: add units.
- Please, use the same verb form throughout the manuscript. For instance, in line 109 you are using the past ("the P-M formula was used") while in line 170 you are using the present ("t test method is used").
Author Response
Thanks for your useful comments. Please see the attachment.

Reviewer 2 Report
The presented paper is an original article with focus on ET at Tibetan plateu within the time period 1961-2017. Methodology is clearly written, presented results are understandable and conclusions are appropriate to results. I have some recommendations/questions/suggestions:
- The lines 69-81 (Introduction), please to move into Discussion
- Canopy reflectance in eq.3: You used the value of 0.23, please to explain why do you use this value (please to cite the source). Canopy reflectance has a seasonal variation due to changes in vegetation cover. You could to use may be more appropriate value for each meteo stations e.g. from remote sensing data of canopy relectance. Please to explain and clarify, or discuss.
- Fig.2 - (a) please to add the legend for y-variable, (b) please explain the parameters UFk, UDk in the text
- Chap 3.4 - Spearman analysis evaluate the relationships between independant and dependant variables. But, e.g. Tmean, Rn are not independant variables due to equation 1. There is an autocorrelation relationship. Please to explain.
Author Response

(The authors gave the same response as above.)

Reviewer 3 Report
The Authors introduce the notion of ‘evaporation paradox’ almost at the end of the paper (in the ‘Discussion’ section). The Reviewer suggests that a (slightly) more detailed description of such phenomenon should be provided at the beginning of the paper (within the ‘Introduction’ section), in order to make the manuscript more comprehensible to the reader.
Author Response

(The authors gave the same response as above.)

Reviewer 4 Report
The reviewer wants to thank the authors for their paper presenting an investigation of the reference evapotranspiration for the Tibetan Plateau. S/he has some (small) comments/questions/suggestions:
*1) General starting with Line (L) 43: Please use Guo et al. [20] instead of Guo, XL et al. and no further reference of the literature. Please correct this for the complete paper. Thank you.
*2) L 92 and ongoing: Why are the Figures marked as a link but not used as such?
*3) L101: Are all 46 stations build in the same year (1961)? Are they all similar equipped with the same instruments? Please clarify also the time resolutions which is available for each measured value.
*4) How did the author react on missing data points and potentially damaged stations? How if the instrumentation was changed, which might happen multiple times in close to 60 years.
*5) L102: The reviewer would like to have here some more information, why only this specific time series from 1961 to 2017 was used. We now have the end of 2020, so at least two more years should be available.
*6) L 200 and ongoing: Please integrate those figures direct in the text. It will only expand the paper in an insignificant way. The original data and the ETo-values should be provided there instead, which allows to have a look at the results.
*7) L327: This paints the picture that the only novelty of this paper is a longer time series. Please clarify this and also clearly show the novelty of the paper in the conclusion.
The reviewer is looking forward to reading the paper again. Thank you.
Author Response

(The authors gave the same response as above.)

Round 2
Reviewer 4 Report
The reviewer wants to thank the authors for their corrections/clarifications of the paper as well as the answers. S/he is looking forward to the publication.